# Is the Blood an Alternative for Programmed Cell Death Ligand 1 Assessment in Non-Small Cell Lung Cancer?

**DOI:** 10.3390/cancers11070920

**Published:** 2019-06-30

**Authors:** Emmanuel Acheampong, Isaac Spencer, Weitao Lin, Melanie Ziman, Michael Millward, Elin Gray

**Affiliations:** 1School of Medical and Health Sciences, Edith Cowan University, 270 Joondalup Drive, Joondalup, WA 6027, Australia; 2School of Biomedical Sciences, University of Western Australia, 35 Stirling Highway, Crawley, WA 6009, Australia; 3Department of Medical Oncology, Sir Charles Gairdner Hospital, Hospital Avenue, Nedlands, WA 6009, Australia; 4School of Medicine, University of Western Australia, 35 Stirling Highway, Crawley, WA 6009, Australia

**Keywords:** PD-L1, non-small cell lung cancer, circulating tumour cells, immunotherapy

## Abstract

Anti-programmed cell death (PD)-1/PD-ligand 1 (L1) therapies have significantly improved the outcomes for non-small cell lung cancer (NSCLC) patients in recent years. These therapies work by reactivating the immune system and enabling it to target cancer cells once more. There is a general agreement that expression of PD-L1 on tumour cells predicts the therapeutic response to PD-1/PD-L1 inhibitors in NSCLC. Hence, immunohistochemical staining of tumour tissue biopsies from NSCLC patients with PD-L1 antibodies is the current standard used to aid selection of patients for treatment with anti-PD-1 as first line therapy. However, issues of small tissue samples, tissue heterogeneity, the emergence of new metastatic sites, and dynamic changes in the expression of PD-L1 may influence PD-L1 status during disease evolution. Re-biopsy would expose patients to the risk of complications and tardy results. Analysis of PD-L1 expression on circulating tumour cells (CTCs) may provide an accessible and non-invasive means to select patients for anti-PD-1 therapies. Additionally, CTCs could potentially provide a useful biomarker in their own right. Several published studies have assessed PD-L1 expression on CTCs from NSCLC patients. Overall, analysis of PD-L1 on CTCs is feasible and could be detected prior to and after frontline therapy. However, there is no evidence on whether PD-L1 expression on CTCs could predict the response to anti-PD-1/PD-L1 treatment. This review examines the challenges that need to be addressed to demonstrate the clinical validity of PD-L1 analysis in CTCs as a biomarker capable of predicting the response to immune checkpoint blockade.

## 1. Introduction

Lung cancer incidence rates have increased substantially since the start of the 20th century, with non-small cell lung cancer (NSCLC) accounting for approximately 85% of reported cases [1]. Survival for advanced NSCLC has improved dramatically with the advent of molecular targeted therapies such as erlotinib and crizotinib for the treatment of patients whose tumours harbour specific sensitising mutations in the epidermal growth factor receptor (EGFR) and translocations in the anaplastic lymphoma kinase (ALK) [2,3]. About 10–15% of Caucasians with lung adenocarcinoma harbour EGFR mutations, generally in never smokers [4], while ALK rearrangements have been described in almost 4–7% of lung cancers, principally among light and non-smokers [5].

The introduction of immune checkpoint inhibitors targeting the programmed cell death-1 (PD-1) and programmed cell death ligand-1 (PD-L1) have shown durable clinical responses with manageable toxicity [6]. PD-1 is expressed on T cells, B cells, and myeloid cells and has two ligands by which it can be stimulated, PD-L1 and PD-L2. Both PD-L1 and PD-L2 can be found in a plethora of different tissues including lymphoid, heart, and lung tissues [7]. The gross result of the interaction between PD-1 and PD-L1 is the inhibition of T cell activation and proliferation, the blockage of production of cytokines, and the general obstruction of immune processes that would otherwise lead to the destruction the target cell [8]. As PD-L1 is expressed in both lymphoid and non-lymphoid tissues, it is thought that the interaction between PD-1 and PD-L1 regulates immune responses in both secondary lymphoid organs and target organs [9]. Tumour cells and other tumour microenvironment cells can express high levels of PD-L1 that can bind to PD-1 on activated T-cells, thereby deactivating the anti-cancer cytotoxic T-cell response [10,11]. Antibodies blocking PD-1 interaction PD-L1 can reverse this process, augmenting the T-cell cytotoxicity and controlling tumour growth.

Anti-PD-1/PD-L1 antibodies have shown superiority to chemotherapy in several NSCLC clinical trials [12,13,14]. This was followed by United States Food and Drug Administration (FDA) approval of four immune checkpoint inhibitors to treat NSCLC; namely, pembrolizumab, nivolumab, atezolizumab, and durvalumab, which are indicated for first or subsequent treatment lines [15,16]. Atezolizumab, nivolumab, and durvalumab indications are independent of PD-L1 expression, while pembrolizumab is restricted to patients whose tumours do not have EGFR or ALK genomic aberrations and express PD-L1 with a tumour proportion score (TPS) ≥1% determined by an FDA-approved test.

Approval of nivolumab was based on reports from two phase III trials that evaluated nivolumab versus docetaxel, one for advanced squamous-cell NSCLC and the other for nonsquamous-cell NSCLC [12,13]. Brahmer et al. (CheckMate 017) reported that the objective response rate with nivolumab was significantly higher compared with docetaxel; however, tumour PD-L1 was not predictive or prognostic of efficacy, regardless of the levels of PD-L1 [12]. Conversely, CheckMate 057 found a predictive association of PD-L1 expression with clinical benefit from nivolumab treatment PD-L1 in patients with advanced non-squamous NSCLC after failure of platinum-based chemotherapy [13]. Similarly, atezolizumab was approved based on the phase III of the OAK trial [17]. The authors reported that NSCLC patient with tumours expressing high PD-L1 levels (TC3 or IC3) experienced the greatest clinical benefit from atezolizumab treatment rather than docetaxel. Patient with less than 1% PD-L1 expression (TC0 and IC0 subgroups) also had improved overall survival [17]. Durvalumab was approved by FDA to treat stage III, unresectable, locally advanced NCSLC that has not progressed after concurrent chemotherapy, based on the PACIFIC trial results [18]. Patients enrolled in the PACIFIC Trial derived a progression free survival (PFS) benefit with durvalumab regardless of the level of PD-L1 expression before chemotherapy.

First-line pembrolizumab monotherapy was approved for stages IIB–IV NSCLC with ≥50% of PD-L1 expression and without EGFR mutation, or ALK or ROS1 rearrangement based on data from two clinical trials, KEYNOTE-010 and KEYNOTE-024, conducted in NSCLC patients with positive PD-L1 expression [14,19]. In the phase I trial, KEYNOTE-001, improved clinical activity was observed in patients who expressed PD-L1 in at least 50% of tumour cells [20], leading to the recruitment of patients with PD-L1-positive tumours (≥1%) in subsequent trials. In KEYNOTE-010, patients expressing positive PD-L1 with a tumour proportion score ≥50% had a favourable hazard ratio (HR) for overall survival (OS) and PFS derived benefit for pembrolizumab over doxetaxel [19]. Congruently, there was a significantly longer PFS and OS benefit with pembrolizumab compared with standard platinum-based chemotherapy in the KEYNOTE-O24 trial that enrolled only NSCLC patients with ≥50% PD-L1 tumour proportion score [14]. Recently, the FDA expanded the indication of pembrolizumab to be included in frontline treatment of NSCLC patients with a PD-L1 expression level ≥1% that do not harbor ALK or EGFR mutations, who are unqualified for surgery or conclusive chemoradiation based on findings from the phase III KEYNOTE-042 trial [21]. The results demonstrated that OS was significantly associated with higher PD-L1 expression levels: TPS ≥ 50% and TPS ≥ 20%; however, patients with a tumour proportion score ≥1% for PD-L1 expression recorded a favourable hazard ratio for OS with frontline pembrolizumab compared with the standard therapy [21]. In addition, based on the promising results from the KEYNOTE-189 trial in 2018, pembrolizumab was added to standard chemotherapy, which includes carboplatin-pemetrexed chemotherapy for nonsquamous NSCLC patients independent of PD-L1 expression [22]. Patients derived benefit for pembrolizumab-combination for OS and PFS across all PD-L1 subgroups, which include a PD-L1 tumour proportion score of <1%, 1–49%, and ≥50%, respectively [22]. Overall, albeit the contradictory results, expression of PD-L1 has demonstrated some utility as a predictor of response to PD-1/PD-L1 inhibitors, and to date is the only biomarker of response utilised in clinic.

Immune checkpoint inhibitors are associated with significant adverse events and immune-related adverse events [6,14,23,24,25,26]. This, combined with the high cost of treatment, underscores the need to find biomarkers of response to treatment. Many predictive biomarkers of response to immunotherapy have been reported in the literature [14,27,28,29]; however, the only biomarker currently used clinically is PD-L1 expression in tumour tissue [30]. There are several problems associated with utilising PD-L1 expression in tumour tissue as a biomarker including, but not limited to scoring discrepancies, different antibody clones approved for different companion diagnostics, intra and inter tumour heterogeneity, and the risk of complications and delayed results associated with re-biopsy. In addition, PD-L1 expression is not always assessed in a contemporaneous tumour to when treatment decisions need to be made [20,30,31,32].

Circulating tumour cells (CTCs) have emerged as a potential alternative tumour sample to evaluate PD-L1 expression. CTCs have been demonstrated to express PD-L1 [33,34,35,36,37,38,39,40]. Moreover, they are expected to express the same immune escape mechanisms as the tumour that they arose from and represent the sum of all tumours in a patient at the time of collection (Figure 1). This has led to several studies assessing PD-L1 expression on NSCLC derived CTCs to determine its potential as a biomarker.

In this review, we summarise studies conducted to assess the prevalence of PD-L1 expression on CTCs from patients with metastatic/non-metastatic NSCLC, evaluate the prognostic value of PD-L1 positive CTCs to inform risk stratification, and explore its utility as a predictive biomarker for selecting patients for immunotherapy.

## 2. Circulating Tumour Cells in NSCLC

Some tumour cells can enter the bloodstream via intravasation. These cells may then exit the bloodstream at a distant capillary bed via extravasation and replicate to form a second tumour. While these cells are in the bloodstream, they are termed circulating tumour cells or CTCs [41]. CTCs may be shed from both the primary tumour and/or metastatic lesions and dispersed into the bloodstream [42,43]. In NSCLC, CTCs, once isolated, are distinguished from other cells in the blood by various means, including the expression of the folate receptor transcript, gene membrane array-assays, and their expression of epithelial markers such as cytokeratin (CK) and epithelial cell adhesion molecules (EpCAM) [44,45,46]. Candidate CTCs must also be negative for the expression of CD45, a marker specific to white blood cells [44].

Several well-established methods for the isolation and detection of NSCLC CTCs have been published, with detection rates varying between 40% and 100% with differing levels of specificity and sensitivity [47,48]. CellSearch, a positive immunoaffinity enrichment method for the isolation of EpCAM expressing cells, has been most commonly used in the clinical setting; however, several studies have used different isolation methods. These include positive or negative immunoselection, size-based filtration, density-based filtration, microfluidic devices, electrophoresis, and immunomagnetic isolation. The advantages and caveats of these collective methods have been reviewed previously [45,49,50,51,52].

The clinical validity of CTC detection in NSCLC patients has been discussed in multiple reviews [45,49,53]. Particularly, studies have underscored the potential for CTC detection for early diagnosis of NSCLC [45,49,54], as well as the prognostic potential of CTCs in early and late stage NSCLC [45,52,53]. The prognostic role of CTCs has been further explored in clinical trials of targeted agents in NSCLC [55]. However, these studies utilised different CTC number cut offs, isolation methods, patient populations, and treatment regimes, thus making it difficult to draw overarching conclusions on the prognostic and predictive roles of CTCs in NSCLC [52]. Nonetheless, the reports from many of the reviewed studies are concordant in suggesting an association between high CTC count and poor prognosis in both early and advanced NSCLC.

## 3. Expression of PD-L1 on CTCs in NSCLC

A total of nine published studies were identified addressing the expression of PD-L1 on CTCs. These were found after searching for CTC studies and PD-L1 expression in National Center for Biotechnology Information (NCBI) PubMed using both circulating tumour cells and PD-L1 related terms together (‘CTC’, ‘programmed death ligand 1′, or ‘programmed cell death ligand 1′), disease terms (‘lung cancer or non-small cell lung cancer’ or ‘NSCLC’), and combined terms. These selected publications focused on late-stage/metastatic NSCLC and assessed the associations between PD-L1 expression on CTCs, prognosis, and/or response to PD-1/PD-L1 inhibitors (Table 1).

Among these nine publications, the number of patients with CTCs (referred herein as CTC detection rate) varied between 22% and 100% at both baseline and after front-line therapy. Several different methods were employed for the isolation and detection of CTCs, making it difficult to compare the results between studies. Three studies employed the size of epithelial tumour cells (ISET) platform [36,37,39], while all other studies employed different types of methodologies such as the CellSearch System [33], the Epic CTC platform [35], Vortex high throughput (HT) technology [40], graphene oxide (GO) Chip [38], spiral microfluidic technology [56], and the CellSieve Microfiltration Assay [34] (Table 1). Despite differences in CTC detection rates and methodologies, these studies indicate that the detection of CTCs provides potential for predictive assessment in advanced NSCLC, in agreement with previous literature [52].

All studies reported on the presence of PD-L1 expressing CTCs in NSCLC patients, with detection rates ranging between 2.0% and 96.8%. Three studies evaluated PD-L1^+^ CTC in patients undergoing anti-PD-1 therapy [33,36,40], with only two of them evaluating whether PD-L1 expression on CTCs can serve as a predictive biomarker of response [36,40]. Four studies evaluated the dynamics of PD-L1 expression on CTCs prior to treatment and after treatment initiation [33,34,37]. However, only two studies compared the expression of PD-L1 on CTCs with that on matching tumour tissue [36,39]. All but one study was single centered, conducted in non-metastatic NSCLC patients with cohort sizes varying from a total of 13 to 112 NSCLC patients (Table 1).

Boffa et al. described the only multi-institutional prospective study reported to date evaluating PD-L1 expression on CTCs in 112 treatment naïve NSCLC patients, the largest sample size analysed to date. PD-L1^+^ CTCs were detectable in the peripheral blood of 23% of NSCLC patients assessed prior to therapy. While most of the circulating cells identified in this study met the consensus criteria for CTCs—expression of epithelial protein, absence of CD45 expression, and an intact nucleus—many PD-L1^+^ CD45^−^ cells in patient samples contained both a nuclear morphology distinct from surrounding white blood cells and other CTCs and with expression of CK below the analytical cut-offs of the assay. These cells were not observed in healthy controls and have not been genetically confirmed to be of malignant origin; therefore, the authors refrain from labelling them as “CTCs” and adopted the nomenclature circulating cells associated with malignancy (CCAMs). Within the PD-L1^+^ CCAM population (47 cells from 26 NSCLC patients), two distinct subpopulations were noted based on differential expression of cytokeratin (CK). Nineteen (40%) cells were positive for CK [PD-L1^+^ CK^+^], whereas 60% were negative for CK [PD-L1^+^ CK^−^].

Ilie et al. investigated the utility of CTCs as a non-invasive biomarker to evaluate PD-L1 status in 106 advanced NSCLC patients. CTCs were detected in 75% of patients. PD-L1 staining was assessed in 71 samples that showed >1 CTCs, but only 8% of NSCLC patients exhibited PD-L1^+^ CTCs [39].

Adams et al. tracked PD-L1 expression in circulating tumour and stromal cells in 41 stage I–IV NSCLC patients undergoing radiotherapy. CTCs were identified in 85% and 100% of patient samples prior to (T0) and after radiotherapy (T1), respectively. Three different patterns of the expression of PD-L1 between T0 and T1 were observed in 35 patients who were assessable for both time points. Eleven patients (32%) showed a rise in PD-L1 expression scores from T0 to T1, 18 patients (51%) had no/low PD-L1 expression, and 6 patients (17%) had persistently medium/high PD-L1 at the two-time points. Of note, the authors characterised the two most common CTC subtypes undergoing epithelial-to-mesenchymal transition (EMTCTC); the prognostically relevant pathological definable CTCs (PDCTC); and a subtype of circulating stromal cells, cancer-associated macrophage-like cells (CAMLs). CAMLs were the most prevalent at both times points (at baseline and after induction of treatment) followed by EMTCTCs and PDTCTCs. Both CAMLs and EMTCTCs have been well proven as cancer-specific biomarkers; therefore, data obtained from the study demonstrated that combining CAMLs and EMTCTCs significantly expands the capacity to characterise cellular biomarkers such as PD-L1 in blood-based diagnosis [34].

Guibert et al. monitored the expression of PD-L1 on CTCs among 96 NSCLC patients treated with nivolumab. CTCs were detected in 93% of patients’ samples at baseline, and they found that 83% of the patients had PD-L1^+^ CTCs on at least 1% of CTCs prior to therapy. At the time of progression, 95% of analysed CTC-positive patients had PD-L1^+^ CTCs, of whom 83% had more than 10% of CTCs as PD-L1^+^, compared with 68% of pre-treatment CTCs [36].

Dhar et al. evaluated PD-L1 expression on CTCs among patients receiving nivolumab therapy prior to or during treatment. The study evaluated CTCs from 31 samples collected from 22 patients. They observed that 96.8% of the patient samples had at least 1 CTC and 48.4% of samples had at least 10 CTCs. Using a cut-off criterion of 1.32 CTCs/mL of blood, based on the analysis of 10 healthy controls, they concluded that 14 of 31 patient samples were positive for CTCs. Among patient samples with CTCs, 85.7% had one or more PD-L1^+^ CTCs [40].

Nicolazzo et al. evaluated the presence of PD-L1^+^ CTCs longitudinally, at three-month intervals during therapy in 24 metastatic NSCLC patients. All patients analysed underwent anti-PD-1 therapy through an expanded access program for nivolumab. CTCs were detected in 83% patients and PD-L1^+^ CTCs were detected in 19 of 20 patients with detectable CTCs at baseline. Ten out of 15 patients with detectable CTCs after three months from starting therapy had PD-L1 expressing CTCs, and 5 of 10 patients with detectable CTCs after six months of treatment had PD-L1^+^ CTCs. It was observed that both baseline and at three months after treatment initiation, almost all CTCs expressed PD-L1, irrespective of clinic-pathological characteristics [33].

Kallergi et al. investigated PD-L1 expression on CTCs isolated from 30 NSCLC patients treated with chemotherapy. CTCs were evaluated by Giemsa staining and immunofluorescence (IF). Giemsa staining revealed that 28 of 30 patients and 8 of 11 patients had detectable CTCs before and after their third chemotherapy cycle, respectively, while IF could detect CTCs in 17 of 30 (56.7%) chemo-naïve patients and in 8 of 11 patients after the third chemotherapy cycle. The rate of PD-L1^+^ CTC detection in the group of patients was 26.7% at baseline and 45% after three cycles of front-line chemotherapy in the whole patient group [37].

Wang et al. monitored the dynamic changes of the expression of PD-L1 in CTC in 13 non-metastatic NSCLC patients treated with radiotherapy or chemoradiation. CTCs were detected in all patients throughout the course of treatment and PD-L1^+^ CTCs were found in 25 out of 38 samples collected with an average of 4.5 cells/mL. All patients who received radiation only had significantly increased numbers of PD-L1^+^ CTCs during treatment and, similarly, 7 out of 8 non-metastatic NSCLC patients showed increased numbers of PD-L1^+^ CTCs with concurrent chemotherapy [38].

Kulasinghe et al. phenotypically characterised CTCs in 35 stage IV NSCLC patients for clinically actionable targets. The authors identified CTCs or CTC clusters in 74% of the patients. Using the presence of at least one PD-L1 positive cell as the cut-off for PD-L1 positivity, 10 out of 18 NSCLC patients expressed PD-L1 in CTCs [56].

Altogether, these studies demonstrate that analysis of PD-L1 on CTCs is feasible and could be detected prior to and after frontline therapy. We should underscore that two studies [34,38] reported that PD-L1 expression is significantly increased in patients after radiotherapy, which is consistent with reports from preclinical studies [57,58].

## 4. Prognostic Index of PD-L1 Expression on CTCs in NSCLC

Several studies have reported that PD-L1 expression in tumour tissue is associated with either shorter or longer survival time in NSCLC patients [59,60,61,62]. The prognostic value of PD-L1^+^ CTCs in NSCLC was analysed in five of the nine studies identified (Table 1) [34,35,37,38,39]. Only two studies demonstrate a statistically significant difference in survival probabilities between patients with PD-L1^+^ CTCs [35,38].

Boffa et al. reported that NSCLC patients with >1.1 PD-L1^+^ CTCs/mL of blood at diagnosis had worse two-year survival compared with those with ≤1.1 PD-L1^+^ CTCs/mL (31.2% vs. 78.8%, *p* = 0.002). A multivariable Cox proportional hazard model controlling for staging indicated that the number of PD-L1^+^ CTCs/mL is a significant independent predictor of mortality (HR = 3.85; 95% CI, 1.64–9.09; *p* = 0.002) [35]. On the other hand, Wang et al. showed that patients with >5% PD-L1 expression on CTCs had significantly shorter PFS compared with PD-L1 negative patients (median 7.1 months vs. median not reached: *p =* 0.017). However, no Cox regression analysis was performed nor multivariate analyses to control for other predictors of progression [38].

It is important to note that none of the four studies analysed patients treated with anti-PD-1/PD-L1 therapy.

## 5. Correlation Between PD-L1 Expression on Tumour Biopsy Tissue and CTCs

Four studies explored the correlation between the expression of PD-L1 on CTCs and its expression in tumour tissue biopsies [34,36,39,40]. The largest reported study to date (*n* = 71) was conducted by Ilié et al., who noted 93% concordance between PD-L1 expression on CTCs and matched tumour tissues [39]. In contrast, Guibert et al. found no statistically significant correlation between the expression of PD-L1 on archived tissue and CTCs (*n* = 66), with the observed rate of concordance being 45% [36]. The other two studies only presented descriptive data indicating some concordance between PD-L1 expression on CTCs and matched tumour tissue. However, they had a limited sample size (*n* = 9 and *n* = 4), which prevented appropriate statistical analysis from being performed [34,40].

Generally, these studies demonstrated the feasibility of comparing PD-L1 expression on CTCs and matched tumours; however, the findings from these studies are not directly comparable because of the different anti-PD-L1 monoclonal antibodies used between CTCs and matched tumours, and different antibodies used between studies. Adams et al. used three anti-PD-L1 monoclonal antibodies for staining, with clone 130021 (R&D Systems, Minneapolis, MN, USA) being used for CTCs, while matched tumours were stained with either the anti-PD-L1 clone 28.8 (DAKO) or the clone 22C3 (DAKO). Similarly, Dhar et al. used two different antibodies, with an anti-PD-L1 antibody (ProSci Inc Ref# 4059, Poway, CA, USA) being used to stain CTCs and the anti-PD-L1 clone SP142 (Ventana) being used for matched tissues. Ilié et al. used an anti-PD-L1 monoclonal antibody clone SP142 for PD-L1 staining both on CTCs and matched tumour tissue, while Guibert et al. used the anti-PD-L1 rabbit monoclonal antibody clone E1L3N (Cell Signalling Technology, Danvers, MS, USA). The use of different antibodies between studies may explain the discrepancies in results.

Four different PD-L1 immunohistochemistry (IHC) assays (PD-L1 assays (22C3, 28-8, SP142, SP263) have been approved by the United States Food and Drug Administration as companion diagnostic tests for tissue staining. Reports from phase I of the blueprint study (BPI) by Hirsh et al., 2017 revealed that three of the four antibodies (22C3, 28-8, SP263) demonstrate similar analytical performance for tumour cell staining, whereas the fourth (SP142) provides significantly lower staining for tumour proportion score [63]. Recently, results obtained from phase II of the blueprint PD-L1 IHC assay (BP2) study using real-life clinical lung cancer samples affirmed the result of BP1 and consolidated the evidence for interchangeability of three different assays (22C3, 28-8, and SP263) [64]. Compared with the above-mentioned studies, these findings demonstrate that the best approach for comparing PD-L1 IHC testing is to perform the test with assays that have undergone rigorous analytical and clinical validation. Similar studies will be needed to develop standardised methods for PD-L1 assessment on CTCs.

## 6. PD-L1 Expression on CTCs as a Predictive Marker in NSCLC

Four of the studies highlighted here explored PD-L1 expression on CTCs in association with the response to PD-1 inhibitors [33,36,38,40]. Only Guibert et al. correlated PD-L1^+^ CTCs with the response to PD-1 inhibitor, nivolumab in a large cohort of NSCLC patients (*n* = 96). The presence of PD-L1^+^ CTCs prior to treatment at different thresholds had no statistically significant impact on PFS. Opposite to what would be expected, patients with >1% PD-L1^+^ CTCs at baseline were frequently non-responders compared with patients who had PD-L1^−^ CTCs (47/69 (68%) vs. 6/15 (40%), *p* = 0.04) [36]. Similarly, Dhar et al. analysed the association between PD-L1 expression and PFS in 17 patients prior to starting treatment. They found that having ≥2 PD-L1^+^ CTCs was not a predictor of response or clinical benefit, though this small cohort included patients treated with nivolumab as well as pembrolizumab [40]. Nicolazzo et al. did not specifically address the predictive significance of PD-L1 in CTCs in their study. Nevertheless, PD-L1^+^ CTCs were identified in 19 patients at baseline, and only 5 of them (26%) achieved clinical benefit after six months of treatment. Thus, PD-L1^+^ CTCs were not demonstrated to be a predictor of response to therapy [33].

Overall, no report to date has shown evidence on the predictive value for PD-L1^+^ CTCs for response to anti-PD1 agents.

## 7. Challenges Associated with PD-L1 Expression on CTCs

The analysis of PD-L1 expression on CTCs is a potential biomarker that can be evaluated with minimal invasion and can be used to assess the dynamic changes of PD-L1 expression on CTCs during treatment. However, the data presented in the studies published so far provide limited information on the clinical validity and utility of PD-L1 analysis on CTCs in patients with NSCLC. The limitations of these studies include small patient cohorts, use of different assays to detect PD-L1, a lack of clear cut-offs for defining PD-L1 positivity on CTCs, and the inability to demonstrate concordance between PD-L1 expression on CTCs and tumour biopsy tissue.

The staining pattern observed in CTCs is one such challenge, with membranous staining of PD-L1 in tumour cells being a standard for interpretation of tissue staining [32]. However, some CTCs exhibited cytoplasmic staining with or without membranous staining [39]. More troublesome, is the choice of antibody utilised for PD-L1 detection. The studies reviewed in here used different antibody clones for the PD-L1 staining of CTCs, some of which are not the FDA-approved antibodies for PD-L1 staining in tumours. Only Kulasinghe et al. and Ilié et al. used FDA-approved anti-PD-L1 antibodies, 28-8 and SP142 [39], with SP142 PD-L1 previously confirmed to be a clear outlier as it detected significantly less tumour cell PD-L1 expression [63,64]. Thus, it remains to be evaluated whether the use of FDA-approved anti-PD-L1 antibody clones may yield different results in future CTC studies.

Another challenge is the differences in detection rates and the number of CTCs between studies. There are a variety of potential factors affecting the CTC enrichment numbers including, but not limited to the use of different CTC isolation and detection methods, biased and highly selective patient cohorts, the time delay between blood collection and analysis, and different treatment modalities. Isolation methods differed largely between studies, with the ISET platform being the most frequently used. This could, in part, be because of the previous chemotherapeutic treatment, NSCLC subtypes, tumour size, and stage, or because of the fundamental issue of tumour heterogeneity both within and between tumours and patients.

The limitations discussed above underscore the need for future studies addressing several critical questions, as reviewed in Table 2, with the ultimate aim to ascertain if CTCs could indeed serve as a non-invasive predicted biomarker to triage patients into anti-PD1/PD-L1 treatment.

## 8. Conclusions and Future Perspectives

The studies reviewed here demonstrated that PD-L1 expression could be assessed on CTCs and that their presence is associated with worse survival outcomes, in the absence of immune checkpoint blockade inhibition. Only two studies to date have evaluated the predictive value of PD-L1 expression on CTCs, with major constraining limitations.

The challenges associated with PD-L1 IHC assays on tissue have led researchers to adapt liquid biopsies to determine tumour PD-L1 expression using CTCs. Assessing PD-L1 expression on CTCs could potentially serve as a surrogate for the evaluation of PD-L1 in tumour tissue biopsies, which is currently used to stratify patients for anti-PD-1/PD-L1 immunotherapy. This is especially important for second-line therapy, where repeated biopsies are impractical at the time of treatment decision. We highlighted the multiple limitations regarding methodology standardization for evaluation of PD-L1 on CTCs. Finally, large multicentre studies will be needed to demonstrate the clinical utility of PD-L1 assessment on CTCs, and before they can influence clinical practice.

## Figures and Tables

**Figure 1 cancers-11-00920-f001:**
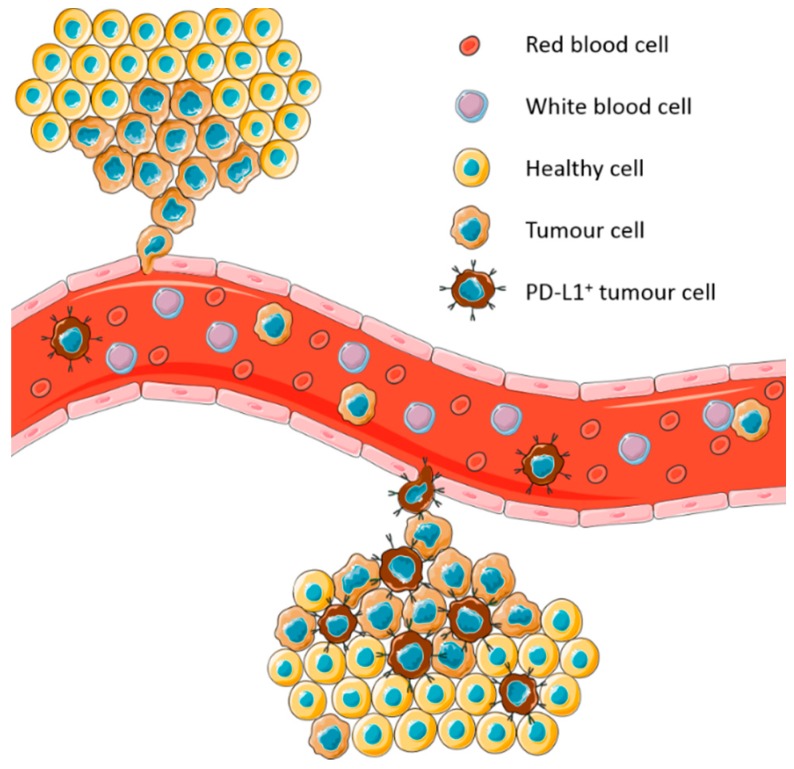
Model of expression of programmed cell death ligand-1 (PD-L1) on circulating tumour cells (CTCs) derived from heterogeneous tumours. CTCs represent the sum of all tumours in a patient at the time of blood collection.

**Table 1 cancers-11-00920-t001:** Prospective studies evaluating programmed cell death ligand-1 (PD-L1) on circulating tumour cells (CTCs) in NSCLC.

Reference (Sample Size)	Detection Method	Therapy	Stage	Tumour Tested for PD-L1 Expression	Time Point of Blood Draw/ Method of Staining	Criteria for CTC Identification	Detection Rates of CTCs	Detection Rates of PD-L1^+^ CTCs	Anti-PD-L1 Antibody	Prognostic/Predictive (Cut-Off Value)
Nicolazzo et al., 2016 [33] (*n* = 24)	CellSearch	Nivolumab	Stage IV NSCLC	No	Baseline 3 months 6 months IF	CK^+^/EPCAM^+^/DAPI^+^ and PD-L1^±^	20/24 (83.0%) 10/15 (66.7%) 10/10 100.0%)	19/20 (95.0%) 10/15 (66.7%) 10/10 (100%)	130021 (R&D system)	NE
* Boffa et al., 2017 [35] (*n* = 112)	Epic Sciences CTC detection plates	Treatment-naïve	Stage I–IV NSCLC	No	Treatment naïve IF	CK^+^/CD45^−^/DAPI^+^/PD-L1^+^	112/112 (100%)	26/112 (23.0%)	EIL3N (Cell Signaling Technology)	Prognostic (>1.1 PD-L1^+^ CTC/mL) [HR = 3.85 (1.64–9.09); *p* = 0.002)]
Ilie et al., 2017 [39] (*n* = 106)	ISET platform; Rare cells	99-Chemotherapy naïve and 7-neoadjuvant chemotherapy	Stage III–IV	Yes	Baseline IF	Circulating non-hematological cells with malignant features (CNHC-MF]	80/106 (75%)	6/71 (8.0%)	SP142 (Ventana)	Not Prognostic (no PD-L1^+^ CTC/mL) PFS: HR: 1.452, (0.82–2.58), and stage-adjusted HR: 1.36, (0.77–2.42)]. OS: HR: 1.55, (0.7–3.470), and stage-adjusted HR: 1.42, (0.63–3.31)].
Adams et al., 2017 [34] (*n* = 41)	Cell Sieve Microfiltration Assay	Chemotherapy Radiotherapy	Stage I–IV NSCLC	Yes	Baseline (T0) Post-induction of Radiotherapy (T1) IF	CK^+^/EPCAM^+^/DAPI^+^	35/41 (85.0%) 41/41 (100%)	35/41 (85.0%) 41/41 100%)	130021 (R&D system) for CTCs 22C3 and 28-8 (DAKO pharmdx) for tissue biopsies	Not Prognostic (≥2 PD-L1^+^ CTC/mL) At T0, PFS: HR = 1.8, *p* = 0.305 At T1, PFS: HR = 0.7, *p* = 0.581
Guibert et al., 2018 [36] (*n* = 96)	ISET platform; Rare cells	Nivolumab	Stage IV NSCLC	Yes	Baseline Post cycle 1 IF	DAPI^+^/CD45^−^/Cytomorphometric malignant features	89/96 (93.0%) 23/24 (95.8%)	74/89 (83.0%) 23/23 (100%)	EIL3N (Cell Signaling Technology)	Not Predictive At ≥ 1% PD-L1^+^ CTC/mL; HR = 1.21(0.64–2.27), *p* = 0.55 At ≥ 5% PD-L1^+^ CTC/mL; HR = 1.05(0.59–1.88), *p* = 0.55] At ≥ 5% PD-L1^+^ CTC/mL; HR = 0.75(0.45–1.25), *p* = 0.27
Dhar et al., 2018 [40] (*n* = 21)	Vortex HT Technology	Nivolumab/Pembrolizumab	Stage IV NSCLC	Yes	Baseline IF	CK^+^/DAPI^+^/CD45^−^ or CK^+^/DAP^+^/CD45^−^/cytopathological features	14/31 (45.2%)	12/14 (85.7%)	4059 (ProSci Inc.)	Not predictive (≥2 PD-L1^+^ CTC/mL) PFS: HR = 0.83 (0.24–2.84, *p* = 0.764)
Kallergi et al., 2018 [37] (*n* = 30)	ISET platform; Rare cells	Chemotherapy-naïve	Stage IV NSCLC	No	Baseline Post cycle 1 Giemsa	CK^+^/CD45^−^/PD-L1^+^	28/30 (93.3%) 9/11 (81.8%)	8/30 (26.7%) 5/11 (45.5%)	EH 12.2H7 (Biolegend) B7-H1/PD-L1/CD274 (Novus Biological)	NE
					Baseline Post cycle 1 IF		17/30 (56/7%) 8/11 (72.7%)			
Wang et al., 2019 [38] (*n* = 13)	GO chip	5-radiation 8-chemoradiation	Stage I–III	No	Baseline (visit 1) Visit 2 Visit 3 IF	CK^+^/CD45^−^/DAPI^+^	13/13 (100%) 12/13 (92.3%) 13/13(100%)	6/13 (46.2%) 10/13 (76.9%) 6/13 (46.2%)	329702 (BioLegend)	PFS analysed (≥5% PD-L1^+^ CTC/mL Log rank *p =* 0.017 HR not provided
Kulasinghe et al., 2019 [56] (*n* = 35)	Spiral Microfluidic Technology	Treatment-naïve	Stage III–IV	No	Baseline IF	CK^+^/CD45^−^/DAPI^+^/PD-L1^+^	18/35 (51.4%)	10/18 (55.6%)	28-8 (Abcam)	NE

IF: immunofluorescence, NSCLC: non-small cell lung cancer, HR: hazard ratio, NE: prognostic or predictive significance not evaluated, ISET: size of epithelial tumour cells, EpCAM: epithelial cell adhesion molecules, CK: cytokeratin, PFS: progression free survival. * prospective multi-institutional study.

**Table 2 cancers-11-00920-t002:** Challenges and opportunities for PD-L1 assessment on CTCs.

Specific Area with Challenges	Potential Question for Future Directions
**Longitudinal PD-L1 Analysis**	Dynamic changes of PD-L1 on CTCs at various treatment time points and relative to the tumour biopsy.
	Changes in PD-L1 expression on CTCs relative to response or resistance to therapy.
**Assay Optimisation**	Optimise a diagnostic assay/test for analysing PD-L1 expression on CTCs, with clear performance measurements and internal transferable controls.
	Comparing diagnostic performance of different PD-L1 assays for PD-L1 expression on CTCs (like the blueprint project aiming at inter-assay harmonization for PD-L1 IHC).
**Clinical Question**	The number of CTCs needed to be detected for accurate assessment of PD-L1 expression in the tumour
	The relevance of the percentage (%) of PD-L1^+^ CTCs compared with the total number of CTCs
	Determining whether longitudinal analysis of PD-L1 expression on CTCs has significant predictive utility.
	Evaluating the relationship between cellular PD-L1 expression in the peripheral blood and the efficacy of immunotherapy affecting the PD-1 axis
**Study Design/Population**	Prospective studies in larger multicentre cohorts of NSCLC patients to evaluate clinical validity of PD-L1 expression on CTCs.

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
