# Peer review of "Is the Blood an Alternative for Programmed Cell Death Ligand 1 Assessment in Non-Small Cell Lung Cancer?"

_cancers, 2019, doi:10.3390/cancers11070920_

Round 1

Reviewer 1 Report

This review describes that programmed cell death-1 (PD-1)/programmed cell death ligand-1 (PD-L1) therapies is promising for non-small cell lung cancer (NSCLC). Analysis of PD-L1 expression on circulating tumour cells (CTCs) may be used for selecting patients for anti-PD-1 therapies. Clinical validity of PD-L1 34 analysis in CTCs as a biomarker is discussed. The manuscript well covers this field and is well written.

It is suggested that you add one section to explain PD-1 and PD-L1 including their functions with an illustration.

Author Response

Response: Detailed explanation of the interaction between PD-1 and PD-L1 including their function has been reported in previous studies, which have been referenced in this current manuscript. Notwithstanding, we expanded our description of the interaction between PD-1 and PD-L1 in the introduction section [page 2, line 56-59].

We have included an illustration (Figure 1), to discuss the expression of PD-L1 on circulating tumour cells in line with the focus of this manuscript.

Reviewer 2 Report

This is an interesting and valuable paper.

Addition of any figure summarizing reviewe or the disussed hypothesis will improve the quality of paper.

Author Response

Response: A Figure illustrating the expression of PD-L1 on circulating tumour in the blood stream has been provided in the revised manuscript – Page 3, Figure 1.

Reviewer 3 Report

This manuscript is well organized and it reviewed the studies of the PD-L1 expression on CTCs and  the prognostic value of PD-L1 positive CTCs comprehensively. It was well written and it is qualified to be published.

Author Response

Response: The authors are grateful for the recommendation

Reviewer 4 Report

This is an important review article aimed at doing a literature review to verify for a presence (or absence) of positive correlation between PD-L1 expression in circulating tumor cells (CTCs) and anti-PD-L1-based immune-check point therapy. The article while interesting has some deficiencies (detailed below).

1) The article is largely descriptive. Table 1 is just summarizing the finding from 9 different articles. The authors should have performed a meta-analysis kind statistical analysis on the 9 studies (reported in the Table 1) and detail their pooled finding, reporting as: overall survival hazard ratio or Odd’s ratio etc in a Forest plot format. Further, if the authors have identified any biases in these 9 studies they should point these caveats in the study.

2) While PD-L1 expression is suggested to be predictor of anti-PD-L1 based immune check point therapy. Till now, no randomized study has proved this assertion. Of note, it is important to point out three major studies KEYNOTE-006, OAK and POPLAR trails, all which conclusively suggested that there is no correlation between PDL1 expression and the therapeutic success of these specific drugs. This point should be explicitly mentioned by adding a paragraph either in the introduction or in the section-7 (challenges section).

Author Response

Response: In addition to summarising the findings on these 9 articles, we provided a critical opinion on the limitations and caveats of these studies and highlighted critical points that need to be address before PD-L1 testing on CTCs can be considered for clinical application.

After serious consideration and discussion with our biostatistician, we concluded that it is not appropriate to perform a meta-analysis with the data from these articles because most of them were observational studies with very small sample size and high heterogeneity in the methods utilised and the characteristic of the study cohorts. The variability was associated with treatment agents, PD-L1 antibody used for immunohistochemistry assay, different CTC isolation methods and different criteria for detecting CTCs.

Reviewer 5 Report

The authors present a review of the research focused on utilizing circulating tumor cells as diagnostic markers for assessing the viability of anti-PD1-based immunotherapy for NSCLC patients. They have chosen to focus on investigation of PD-L1 expression in CTCs with the intent to use that as the primary marker instead of tumor biopsies. To this end, they have discussed in detail instances where PD-L1 expression in CTCs have been observed, their correlation to tumor biopsy-based markers and challenges associated with the paradigm. They have also briefly discussed techniques for isolating and characterizing CTCs in the clinic, their utility as a prognostic index and using PD-L1 as a predictive marker. I believe that the authors have covered several important topics within the scope of the review while maintaining a focus on the central subject. Further, this review is well-organized, comprehensive and easy-to-read, which will benefit a broad audience from different backgrounds.

The following are minor issues, I would strongly suggest the authors to address prior to publication:

1.      Line 87: The authors refer to a Figure 1, which is not included in the manuscript

2.      Line 104: The authors repeatedly use the phrase “CTC detection rate”, which should be described for readers not familiar with CTC characterization.

3.      Line 111: Most recently, a size-based microfluidics device has also been developed for isolating CTCs. The authors might want to cite this as well: Eric Lin, Lianette Rivera-Báez, Shamileh Fouladdel, Hyeun Joong Yoon, Stephanie Guthrie, Jacob Wieger, Yadwinder Deol, Evan Keller, Vaibhav Sahai, Diane M. Simeone, Monika L. Burness, Ebrahim Azizi, Max S. Wicha, Sunitha Nagrath. High-Throughput Microfluidic Labyrinth for the Label-free Isolation of Circulating Tumor Cells. Cell Systems, 2017

4.      The authors can have an additional column in Table 1 with the criteria used to identify CTC

Author Response

1.    Line 87: The authors refer to a Figure 1, which is not included in the manuscript

Response: Figure 1 has been included in the revised manuscript

2.    Line 104: The authors repeatedly use the phrase “CTC detection rate”, which should be described for readers not familiar with CTC characterization.

Response: The defined “CTC detection rate” as the percentage of patients with detectable CTCs in page 5 line 169-170.

3.    Line 111: Most recently, a size-based microfluidics device has also been developed for isolating CTCs. The authors might want to cite this as well: Eric Lin, Lianette Rivera-Báez, Shamileh Fouladdel, Hyeun Joong Yoon, Stephanie Guthrie, Jacob Wieger, Yadwinder Deol, Evan Keller, Vaibhav Sahai, Diane M. Simeone, Monika L. Burness, Ebrahim Azizi, Max S. Wicha, Sunitha Nagrath. High-Throughput Microfluidic Labyrinth for the Label-free Isolation of Circulating Tumor Cells. Cell Systems, 2017

Response: As suggested by the reviewer, this article has been cited in the revised manuscripts in page 4 line 151 – ref 51.

4.    The authors can have an additional column in Table 1 with the criteria used to identify CTC

Response: An additional column has been added to Table 1 describing criteria used to identify CTC.

Round 2

Reviewer 4 Report

This is a significantly improved manuscript and should be published. I commend authors for their hardwork and critical analyses.